# The Vps13 Family of Lipid Transporters and Its Role at Membrane Contact Sites

**DOI:** 10.3390/ijms22062905

**Published:** 2021-03-12

**Authors:** Samantha Katarzyna Dziurdzik, Elizabeth Conibear

**Affiliations:** 1Centre for Molecular Medicine and Therapeutics, British Columbia Children’s Hospital Research Institute, University of British Columbia, Vancouver, BC V5Z 4H4, Canada; sdziurdzik@cmmt.ubc.ca; 2Department of Medical Genetics, Faculty of Medicine, University of British Columbia, Vancouver, BC V5Z 4H4, Canada

**Keywords:** Vps13, Atg2, membrane contact sites, lipid transport, yeast model, chorea acanthocytosis, Cohen syndrome, Parkinson’s disease, ataxia

## Abstract

The conserved VPS13 proteins constitute a new family of lipid transporters at membrane contact sites. These large proteins are suspected to bridge membranes and form a direct channel for lipid transport between organelles. Mutations in the 4 human homologs (*VPS13A–D*) are associated with a number of neurological disorders, but little is known about their precise functions or the relevant contact sites affected in disease. In contrast, yeast has a single Vps13 protein which is recruited to multiple organelles and contact sites. The yeast model system has proved useful for studying the function of Vps13 at different organelles and identifying the localization determinants responsible for its membrane targeting. In this review we describe recent advances in our understanding of VPS13 proteins with a focus on yeast research.

## 1. Introduction

The large, evolutionarily conserved VPS13 (vacuolar protein sorting 13) proteins are newly identified lipid transport proteins that localize to membrane contact sites (MCSs), regions where organelle membranes are tethered in close apposition (10–30 nm). The proximity of membranes at these sites enhances the rate and efficiency of lipid transport by allowing lipid transport proteins to directly transfer lipids between membranes in a non-vesicular manner [1,2]. Disruptions of lipid transport at membrane contact sites are associated with a number of neurological disorders, highlighting the importance of organelle lipid homeostasis for neuronal function [3,4,5,6,7,8]. Strikingly, mutations in the genes encoding the 4 homologous VPS13 proteins are associated with distinct neurological disorders: chorea-acanthocytosis (*VPS13A*) [9], Cohen syndrome (*VPS13B*) [10], early onset Parkinson’s disease (*VPS13C*) [11] and ataxia (*VPS13D*) [12,13]. The phenotypic differences associated with mutations in each human *VPS13* homolog are not explained by differential expression as the homologs are present in all tissues. Instead, the different human VPS13 proteins localize to different sets of MCSs (Table 1) [14,15,16,17]. The loss of lipid transport at a specific MCS occupied by one VPS13 homolog has been advanced as an explanation for the diverse neurological disorders resulting from dysfunction of individual VPS13 proteins [18,19].

Understanding how VPS13 proteins are localized to contact sites and function in lipid transport has been a focus of recent research, but the large size and low expression level of the human VPS13 proteins makes them challenging to study. Yeast has proven a useful model organism for identifying the localization determinants and functions of VPS13 proteins due to its single, conserved Vps13 that localizes to multiple organelles and MCSs (Table 2). In this review we will describe recent progress towards understanding the lipid transport function and membrane targeting determinants of VPS13 proteins, with a focus on yeast Vps13.

## 2. Vps13 Belongs to a New Family of Lipid Transport Proteins

Vps13 was first identified in a screen for yeast mutants defective in sorting carboxypeptidase Y (CPY) to vacuoles [34]. Vps13 is now known to be required for a wide range of processes that include transport between Golgi and endosomes, trans-Golgi network homotypic fusion, sporulation, and the rescue of mitochondrial defects in mutants lacking a functional endoplasmic reticulum-mitochondria encounter structure (ERMES) [27,28,29,33], an endoplasmic reticulum (ER)-mitochondrial tethering complex that functions in lipid transport [35,36]. As a large protein that lacks obvious functional domains, its function has been a mystery. While a role in canonical vesicle formation and fusion events has not been ruled out, discoveries over the past few years have shown that yeast Vps13 localizes to MCSs and may be involved in lipid transfer at these sites [27,28].

Evidence that Vps13 transports lipids comes from structural and in vitro studies of purified yeast Vps13 protein fragments. A crystal structure of an *N*-terminal region (amino acids 1-335 of *Chaetomium thermophilum* Vps13), which encompasses the conserved chorein domain (Figure 1A), shows a scoop-like structure with a hydrophobic groove along the interior [15]. Single particle cryo-EM of an extended *N*-terminal region (amino acids 1-1390 of *Chaetomium thermophilum* Vps13) revealed that the hydrophobic groove extends beyond the chorein domain (Figure 1B) [37]. Along with the extended groove, this region exhibits a “basket handle-like” structure of unknown function. In vitro studies with the equivalent fragment from budding yeast Vps13 determined that this domain can bind multiple glycerophospholipids with a preference for phosphatidylcholine (PC) and phosphatidylethanolamine (PE) [15]. When tethered to liposomes, this fragment efficiently transfers glycerophospholipids between them providing evidence that Vps13 is a *bona fide* lipid transport protein [15].

Unlike previously characterized lipid transport domains such as steroidogenic acute regulatory-related lipid transfer (StARkin) or oxysterol-binding protein (OSBP) domains, which form a teacup with lid structure and shuttle individual lipids across membrane contact sites [38,39], Vps13’s extended hydrophobic groove binds multiple lipids at once and thus may function in bulk transport of lipids between organelles [15,37]. This elongated lipid transport structure is thought to bridge the two organelle membranes at a contact site providing a channel for rapid, direct lipid transport. Support for this bridge model came from studying a mutant form of Vps13 in which a restricted ring of hydrophobic residues positioned within the groove was substituted for charged residues to block the movement of lipids. This mutant could bind glycerophospholipids, but was unable to function in sporulation suggesting that lipids are shuttled directly across the groove between organelles [37].

The structure of Vps13, as elucidated from X-ray crystallography, cryo- and negative stain EM, is remarkably similar to that of autophagy-related 2 (Atg2), suggesting Vps13 and Atg2 are members of a new class of lipid transfer proteins [15,33,37,40,41]. Atg2 has similar conserved chorein, APT1 and autophagy-related protein 2 *C*-terminal (ATG2_C) domains, as well as an elongated hydrophobic groove (Figure 1C) [40,41,42]. Atg2 functions in autophagy where it is essential for phagophore expansion due to its role in non-vesicular lipid transport between the ER and phagophore [43,44,45]. Thus, both Atg2 and Vps13 appear to mediate bulk lipid transfer between organelle membranes.

Studies of the Atg2 lipid transport domain, which has been better investigated than that of Vps13, suggests how lipids interact with the hydrophobic groove. A crystal structure of Atg2 bound to PE determined that the hydrophilic head group of lipid molecules contacts positively charged residues on the exterior of the groove, while the hydrophobic tails are in contact with the interior hydrophobic concave face [41]. Interestingly, the Atg2 groove has additional gaps which could allow diverse lipid moieties to enter the lipid transport channel [41], unlike the synaptotagmin-like mitochondrial lipid-binding protein (SMP) domain of Mdm12 which tightly associates with the hydrophobic tails of PE, imparting specificity for this lipid [46]. It will be interesting to see if the human VPS13 homologs preferentially transport certain glycerophospholipids and if other lipid species, such as fatty acids, are trafficked by these proteins.

## 3. Vps13 Has a Variety of Localization Determinants

VPS13 proteins share a number of conserved domains (Figure 1C). A low-resolution single-particle EM structure of the intact Vps13 protein shows an extended rod-like structure with a large loop on one end [33]. The rod portion of Vps13 is approximately 20 nm in length which is sufficient to span the distance between two organelles at MCSs, with putative membrane targeting domains on either end of the rod contacting opposing organelles (Figure 2). In this section we will discuss our current understanding of how each conserved domain contributes to membrane targeting.

### 3.1. The N-Terminus as an ER-Targeting Determinant

The role of the Vps13 *N*-terminal region in targeting to membranes has not been investigated. Atg2 has a similar extended rod structure that connects opposing membranes at contact sites, but lacks the large loop [40,45]. A conserved *N*-terminal helical region within the chorein domain of Atg2 binds directly to the ER [47], the major site of lipid biosynthesis [48]. This region is highly conserved in Vps13 suggesting it may rely on the same conserved helix to localize to the ER or to other negatively charged membranes [33]. In fact, *atg2*Δ autophagy defects are rescued by Atg2 chimeras that substitute the *N*-terminus with that of Vps13 [41], or to a lesser extent the ER-localization signal of Sec71 [41].

Human VPS13A and VPS13C proteins also contain an *N*-terminal chorein domain, but have a FFAT (two phenylalanines in an acidic tract) motif in a loop adjacent to this region that is required for ER recruitment through interactions with the ER MCS proteins, VAP-A and VAP-B [15,16]. A non-canonical phospho-FFAT motif, which interacts with VAP proteins only when phosphorylated, was recently found to target VPS13D to the ER [26]. VPS13B also has a predicted phospho-FFAT but its ER localization has not been studied [49]. It is unclear if the *N*-terminus of Vps13 binds the yeast VAP homologs, but so far no predicted FFAT motifs have been found in this region. Therefore, the identity of the membranes that are recognized by the Vps13 *N*-terminus, and the role of the conserved chorein domain in membrane targeting, remain an open question.

### 3.2. Organelle-Specific Adaptors Target Vps13 to Membranes via a Conserved Motif

Yeast Vps13 localizes to multiple organelles and MCSs (Table 2) and binds different membranes under specific nutrient conditions and developmental stages. In vegetative yeast, Vps13 is found primarily at endosomes with a smaller pool at mitochondria [27,28]. Under glucose-limiting conditions, Vps13 relocalizes to the nuclear ER-vacuole MCS known as the nucleus-vacuole junction (NVJ) [18,27,28]. In contrast, during sporulation Vps13 is recruited to prospore membranes which form around daughter haploid nuclei [29]. If Vps13 resides at MCSs at these organelles, the bridge model shown in Figure 2 predicts that one end of the lipid transport rod must be anchored to a variety of organelle membranes.

Recent work has identified Vps13 adaptor proteins specific to each of the previously described sites that dictate these different localizations. Vps13 is recruited to the prospore membrane by the adaptor Spo71 [29], to mitochondria by Mcp1 [31], and to endosomes and vacuoles by Ypt35 [18]. Each adaptor associates directly with an organelle membrane, either through lipid binding Pleckstrin homology (PH) or Phox homology (PX) domains in the case of Spo71 and Ypt35, or membrane-spanning domains in the case of Mcp1. All three adaptors bind Vps13 through a conserved proline-X-proline (PxP) motif, located in a predicted linker region *N*-terminal to the membrane targeting domains [18]. The different adaptors, or their related PxP motifs, compete for Vps13 binding suggesting they interact with a single site but with different affinities. How Vps13 preferentially recognizes a specific adaptor is unclear; one possible method is through adaptor expression level and/or affinity for Vps13. For example, Spo71 is expressed only during sporulation, and its PxP motif is believed to bind Vps13 more tightly than that of Mcp1 or Ypt35 [18], allowing the cellular pool of Vps13 to be redirected to the growing prospore membrane. Whether adaptor-mediated recruitment is regulated through adaptor expression levels, post-translational modifications, or other means is an active area of investigation. It is also of interest to determine if other PxP motif-containing adaptors are required for recruitment to other organelles, such as peroxisomes [31], or for specific Vps13 functions, such as CPY sorting [50].

### 3.3. Vps13 Adaptor Binding Domain

The PxP motif of organelle adaptors binds to a site within a highly conserved Vps13 domain that consists of a six-repeat sequence composed largely of beta sheets [18]. This domain is referred to as the Vps13 Adaptor Binding (VAB) domain, or alternatively as the WD40/β-propeller domain based on its predicted structural fold [15,50], although the true structure of this domain is unknown. In the WD40/β-propeller model, each of the 6 repeats form 2 blades that could fold into tandem 6-bladed β-propellers [51], or a single 12-bladed β-propeller—a rare and perhaps unlikely structure that could plausibly form the large loop structure seen by single particle EM [33,52]. However, this domain lacks key residues associated with canonical WD40 domains [53] and newer protein structure prediction algorithms suggest that other folds are possible, such as β-sandwiches (S.K.D. and E.C. unpublished observations).

A fragment of the VAB domain containing only repeats 5 and 6 binds adaptors [50] suggesting the PxP motif binds to a single site within this region. Interestingly, a systematic analysis of mutant Vps13 proteins with amino acid substitutions in the invariant asparagine residues present in each VAB repeat found both the first and last of the six repeats were important for adaptor recognition [50]. As the invariant asparagine in the first VAB repeat is important for adaptor binding only in the context of the intact Vps13 protein, it could influence the conformation or accessibility of the binding site.

The adaptor binding and localization function of human VAB domains may be conserved. A mutation in *VPS13D* that causes spastic ataxia (N3521S) [13] alters the conserved asparagine of the 6th VAB domain repeat, and blocks adaptor binding and membrane recruitment when the corresponding mutation is modeled in yeast [50]. Recent work determined that VPS13D is recruited to mitochondria by Miro, and this requires a *C*-terminal region that includes the VAB domain [26]. While further work will be needed to determine the precise role of the VAB domain in the membrane recruitment of VPS13D, a VAB domain-containing fragment of human VPS13C localizes to endolysosomes in cultured cells, suggesting it binds a localization determinant [15].

VAB-dependent localization has not been reported for VPS13A or VPS13B. A fragment of VPS13A containing this domain was cytosolic [15]; however, the relevant adaptors may only be expressed in certain cell types, developmental stages or conditions as was observed for VPS13D [26]. Of interest, a disease-causing *VPS13A* homozygous missense mutation (W2460R) that alters a highly conserved hydrophobic residue within the 6th VAB domain repeat [54] perturbs VPS13A localization to lipid droplets (LDs) [55,56].

The VAB domain could have a conserved role in binding adaptors for at least some VPS13 proteins. However, if the 4 human VPS13 proteins have diverged to localize to different subsets of MCSs they must use different localization determinants. Fine-tuning of adaptor motifs and VAB domain adaptor-binding sites would allow different VPS13 proteins to recognize different adaptors. Indeed, adaptors and adaptor motifs seem to undergo rapid evolutionary changes: the consensus PxP motif used in *S. cerevisiae* is conserved only within the order Saccharomycetales [18] and homologs of the mitochondrial and prospore membrane adaptors Mcp1 and Spo71 are found only in fungi. VPS13B has a divergent VAB domain [50] and instead is localized to the Golgi by a *C*-terminal region that binds the small GTPase Rab6 [57], suggesting some VPS13 proteins may not rely on VAB-adaptor interactions as a primary targeting mechanism.

### 3.4. Other Localization Determinants at the C-Terminus

The protein-binding partners discussed thus far interact with either the Vps13 *N*-terminus via VAP proteins or the *C*-terminus through the VAB domain. Some protein binding sites, such as that of Cdc31 [33], have not been mapped to a specific region of Vps13. There is evidence to suggest that other conserved *C*-terminal regions work cooperatively with the VAB domain for membrane targeting. These domains may bind lipids, protein interactors, or sense membrane curvature. Three other highly conserved regions make critical contributions to localization and each is described below in turn (see Figure 2).

#### 3.4.1. APT1 Domain

The APT1 domain is conserved in both VPS13 and ATG2 proteins and has been shown to interact with specific phospholipids in vitro through PIP strips and liposome binding assays. The APT1 domain in yeast Vps13 binds to PI(3)P, while in human VPS13A this domain binds both PI(3)P and PI(5)P [30,58]. The APT1 domain of yeast Atg2 has a similar phospholipid preference, binding to PI(3)P, and to a lesser extent PI(3,5)P_2_ and PI(4,5)P_2_ in vitro [59]. This domain is not related to known lipid-binding domains and is not sufficient to localize to PI(3)P-rich membranes in vivo. Notably, a causative chorea-acanthocytosis *VPS13A* point mutation (I2771R) in the APT1 domain alters its affinity for PI(3)P in vitro [30]. Modeling the homologous mutation in yeast (I2749R) similarly reduced PI(3)P binding. This mutant form of Vps13 was unable to rescue vacuolar protein sorting defects and drug sensitivities of *vps13*Δ null yeast suggesting that this domain is important for VPS13 protein function.

The VPS13A APT1 interaction with PI(3)P-rich liposomes requires bivalent cations such as calcium or magnesium, while its interaction with PI(5)P-rich liposomes does not [58]. The cations are suggested to bind to the *C*-terminus of the APT1 domain as *C*-terminal truncations result in a loss of requirement for bivalent cations [58]. Interestingly, a *VPS13A* chorea-acanthocytosis mutation causing amino acid substitution in the APT1 domain enhanced interactions with phosphatidylinositol (PI)-rich liposomes in the presence of calcium ions. This altered specificity of the APT1 domain for certain PI species could provide a method of regulating recruitment to specific membranes in response to signalling events. Loss of the ability to respond to such signalling could result in VPS13 protein mislocalization and reduced lipid transport at MCSs.

#### 3.4.2. ATG2_C Domains and Conserved Amphipathic Helices

VPS13 and ATG2 proteins have conserved ATG2_C domains at their *C*-termini. These domains contain two to three predicted amphipathic helices (AHs), structures that contain hydrophobic residues on one face of the helix and polar residues on the other [60]. This unique structure allows AHs to insert into membranes, sense regions of membrane curvature and even deform membranes [61]. This domain has not been studied in yeast Vps13. In human VPS13A and VPS13C, the ATG2_C region is sufficient for localization to LDs and, in VPS13A, to mitochondria as well [15]. Disruption of the hydrophobic face of the predicted amphipathic helix in a VPS13A ATG2_C domain-containing fragment blocked localization to LDs and mitochondria, suggesting the AH is required for membrane recruitment.

The ATG2_C domain of ATG2 proteins has been more extensively studied. In yeast, this domain is required for localization to the preautophagosomal structure (PAS), the nascent site of autophagosome formation, and an Atg2 truncation lacking the ATG2_C domain is defective in autophagic flux [47]. Point mutations that introduce disruptions in the hydrophobic face of the AH in full-length Atg2 abolish recruitment to the PAS and impair autophagy [47]. In human ATG2A, the ATG2_C domain localizes the protein to LDs and to ER-mitochondrial contact sites (MAMs) upon nutrient starvation [44,62,63]. Recruitment to MAMs involves binding to TOM40, an outer mitochondrial membrane protein, and the AH is required for this interaction [63].

#### 3.4.3. PH Domain

The final *C*-terminal domain of VPS13 proteins is the PH domain which was identified through structural homology searches [64,65]. Most PH domains interact with phosphatidylinositol phosphates providing a mechanism for membrane targeting [66]. In vitro, a *C*-terminal Vps13 fragment containing both ATG2_C and PH domains binds PI(4,5)P_2_ [33]. A dimer of the yeast Vps13 PH domain weakly localizes to yeast budnecks, and expression of a mutant Vps13 protein lacking the PH domain is defective in NVJ localization in stationary phase yeast [65] suggesting this domain does not bind strongly to lipids or membranes, but may contribute to membrane recruitment through coincidence detection. The role of the PH domain in human VPS13 proteins has not been studied, but mutations resulting in truncation of the VPS13A PH domain are causative for chorea-acanthocytosis suggesting an important function for this domain [54,67].

## 4. Function of Vps13 at Specific Contact Sites

Although a number of localizations have been determined for both yeast and human VPS13 proteins, the full list of MCSs occupied by each homolog is not clear. The single organelle localizations listed in Table 1 and Table 2 are expected to be just one of two organelles tethered by VPS13 at MCSs. In this section, we will discuss possible organelle pairs tethered by VPS13 proteins and the evidence supporting a function in lipid transport at these sites.

### 4.1. Vps13 at the NVJ

In yeast, Vps13 localizes to NVJs during glucose starvation [27], but its function at this contact site is unclear. The NVJ has been implicated in piecemeal microautophagy of the nucleus and lipid droplet biogenesis [68,69,70,71], but a functional role for Vps13 or its NVJ adaptor Ypt35 in these processes has not been uncovered. The nuclear ER and vacuole are a well-defined pair of organelles that are bridged by Vps13. Loss of the vacuolar adaptor, Ypt35, causes Vps13 to redistribute in the ER suggesting it has an ER-targeting determinant [18]. Conversely, loss of Vps13 causes Ypt35 to become more evenly distributed around the vacuolar membrane. Thus, the interaction between Vps13 and Ypt35 clusters both of these proteins at the NVJ but is not required for the association of these individual proteins with the ER or vacuole, respectively.

### 4.2. Vps13 at Mitochondrial Contact Sites

Vps13 is suggested to function in lipid transport at mitochondria based on the study of spontaneously occurring suppressor mutations that allow Vps13 to compensate for loss of the ER-mitochondrial tether ERMES [27,28]. The ERMES complex consists of 4 subunits, 3 of which have lipid-transporting SMP domains [35,36,46]. Loss of ERMES subunits results in slow growth on fermentable media, abnormal, globular mitochondrial morphology and reduced lipid transport between ER and mitochondria [35].

The slow growth and mitochondrial morphology defects caused by loss of two different ERMES subunits are rescued by dominant mutations in *VPS13* but not by expression of a synthetic ER-mitochondrial tether [27,28,35], suggesting that Vps13 does not simply link membranes but also restores impaired lipid transport. The basis for this suppression is unclear. One hypothesis is that the suppressor mutations alter Vps13 localization, and the loss of Vps13 from some contact sites provides a greater pool available to function elsewhere. Indeed, many of these mutations specifically block the localization of Vps13 to the NVJ in glucose limiting conditions [27,28].

It is clear that Vps13 localization to mitochondrial contact sites is required for rescue [18,31]. ERMES suppression requires that mutant forms of Vps13 bind the mitochondrial adaptor, Mcp1 [18,31]. High levels of Mcp1, which redirect wild type Vps13 from endosomal puncta to the surface of mitochondria, also compensate for ERMES defects. Both the Vps13 suppressor mutations and overexpression of *MCP1* may thus increase the levels of Vps13 available to function at a mitochondrial contact site.

Mitochondria make contacts with many other organelles including the ER, endosome, and vacuole [72]. The ability of Vps13 to rescue ERMES mutants was suggested to stem from a role in lipid transport at vacuole-mitochondria contacts (vCLAMPS), due to a requirement for the vCLAMP component Vps39 [27]. In this model, lipids travel from ER to mitochondria by way of the vacuole, bypassing the ER-mitochondrial contact site. However, using separation-of-function alleles it was recently determined that Vps39’s function in vCLAMP formation is not important for this rescue. Vps39 is also a subunit of the homotypic fusion and vacuole protein sorting (HOPS) tethering complex, and it is Vps39’s role as a HOPS subunit that—together with the other HOPS subunits—is required for suppression of ERMES mutants by dominant *VPS13* alleles [73].

Vps13 is not observed at Vps39-containing vCLAMPs [73] but may form a different type of contact between mitochondria and vacuoles. If so, this would involve a new localization determinant, as the only adaptor known to link Vps13 to vacuoles binds to the same site as Mcp1 and is dispensable for rescue of ERMES mutants [18]. Further work will be needed to understand how dominant suppressor mutations alter Vps13 localization or function.

### 4.3. Vps13 and Prospore Membrane Expansion

Vps13 localizes to the prospore membrane during sporulation where it is required for prospore membrane expansion and closure, and the formation of viable spores [29]. Similar defects in sporulation are seen in mutants lacking the Vps13 prospore membrane adaptor Spo71 [74], or a functional Vps13 VAB domain [50]. The dramatic expansion of the prospore membrane that encircles each prospore nucleus is reminiscent of the Atg2-mediated expansion of the phagophore membrane during autophagy, underlining the need for bulk lipid transport to drive this expansion. The small prospore membranes formed in *vps13*Δ mutants have reduced levels of phosphatidic acid (PA), PI(4)P and PI(4,5)P_2_ suggesting alterations in lipid homeostasis [29].

The opposing organelle at Vps13-prospore membrane contact sites is unknown. One suggested source of lipids for prospore membrane expansion is LDs [75]. LDs accumulate at contacts with growing prospore membranes and these docked LDs decrease in size as prospore membrane expansion proceeds. During meiosis II, levels of TAG decrease while free fatty acids increase suggesting lipolysis is occurring, and severe defects in spore viability are seen in cells lacking LDs [75]. It is thus interesting to hypothesize that Vps13 tethers LDs to growing prospore membranes to provide a lipid source for membrane expansion. As Vps13 is not known to transport free fatty acids, coordinated phospholipid synthesis coupled to the release of fatty acids from LDs could provide a source of transportable lipid at these sites.

A similar mechanism drives autophagosome membrane formation, where activation of fatty acids from LDs and localized phospholipid synthesis is required for efficient phagophore elongation and autophagy [76]. Cells lacking LDs or with defects in lipases or phospholipid synthases exhibit autophagy defects, indicating release of fatty acids from LDs may be required for efficient autophagosome membrane expansion [77,78,79,80]. This suggests an interesting model, where enzymes involved in fatty acid activation and phospholipid synthesis are scaffolded with LDs at sites of membrane expansion to provide a pool of newly formed phospholipids for transport into growing autophagosome membranes. As Atg2 is the major lipid transport protein at these sites [41,45], it is interesting to speculate that Vps13 also scaffolds LDs and lipid biosynthetic enzymes at contact sites to provide a source of glycerophospholipids for transport to growing membranes.

### 4.4. VPS13 at Autophagosomal Membranes

Recently, yeast Vps13 was determined to function in cortical ER-phagy, but not in nuclear ER-phagy or bulk autophagy [32]. In *vps13*Δ yeast the degradation of cortical ER markers is defective in cells treated with rapamycin, a potent inducer of autophagy. In this null mutant, ER membranes accumulate near late endosomes and are not packaged into autophagosomes or delivered to vacuoles [32]. In other organisms, VPS13 homologs have an established role in autophagy. In the slime mold *Dictyostelium discoideum*, mutations in the *VPS13* homolog *tipC* reduce autophagic flux [81]. In *Drosophila melanogaster*, mutation of the *VPS13A*/*VPS13C* homolog causes sensitivity to proteotoxic stress and an accumulation of protein aggregates that is rescued by overexpression of human *VPS13A*, suggesting a defect in delivery of autophagosomal membranes to lysosomes [82].

VPS13A has been shown to function in organelle-specific autophagy in human cell lines. *VPS13A* silencing results in impaired lysosomal degradation with accumulation of autophagic substrates [20], and in mitochondrial fragmentation and reduced mitophagy [16]. Surprisingly, yeast Vps13 is not required for mitophagy and in fact deletion of *VPS13* has been found to upregulate mitophagy in yeast [28,32]. As further evidence for a role of VPS13A in autophagy, erythrocytes from chorea-acanthocytosis patients accumulate autophagosomal cargo proteins. These patient erythrocytes have delayed mitochondrial and lysosomal clearance indicative of impaired autophagy [83]. The abnormal, spiculated morphology of these erythrocytes has also been suggested to result from alterations in lipids within the plasma membrane [84,85]. Further work is required to determine if all VPS13 proteins are involved in autophagy, at what stage in autophagy they function, and which organelle contact sites are relevant for these functions.

### 4.5. VPS13 Proteins at LDs

Although yeast Vps13 has not been found at LDs, there is abundant evidence linking human VPS13 proteins to LD function. Mammalian VPS13A and VPS13C localize to ER-LD contacts, with enhanced LD association seen for VPS13A in cells treated with oleic acid, an inducer of LD formation [15,16]. VPS13A and VPS13C localize to ER-LD contacts through FFAT motifs that bind VAP at the ER, and AHs that target LDs [15]. The conserved AHs in Atg2 similarly localize to LDs [44,62], while its *N*-terminal chorein domain binds directly to ER membranes [47] (see Section 3.1). The chorein and AH domains are conserved in yeast Vps13. Although their role in targeting has not been explored, it is plausible that yeast Vps13 protein could similarly tether ER and LDs at either end of its lipid transport groove.

Studies reporting changes in LD size, number, and motility in *VPS13* and *ATG2* mutants suggest they function at LDs. Loss of *ATG2A* and *ATG2B* increases the number and size of LDs [44], while knockdown of *VPS13A* in mammalian cells, or of *Vps13* in Drosophila, similarly increases LD number [16]. The reduced motility of VPS13A-associated LDs suggests VPS13A has a role in tethering LDs to another organelle. Notably, *VPS13C* is highly expressed in adipose tissue, particularly during adipocyte differentiation where VPS13C localizes to the surface of LDs [86,87]. Knockdown of *VPS13C* during adipocyte differentiation impairs adipogenesis [86], and *VPS13C* silencing in brown adipose tissue reduces LD size [87]. Whether all VPS13 proteins localize to ER-LD contacts and whether localization to these contact sites only occurs under certain conditions are important outstanding questions.

### 4.6. Vps13 at Other Contact Sites

It is likely that yeast Vps13 works at other contact sites that have not been identified. For example, Vps13 has a well-established role in the sorting pathway that transports CPY from the Golgi to the vacuole [34]. Adaptor competition experiments suggest that a PxP motif-containing adaptor is involved in this function, but no such adaptor has been identified for this pathway [50]. Vps13 has also been observed at peroxisomes [31], but has not yet been implicated in lipid transport at this organelle. Alternatively, Vps13’s role in some processes, including trans-Golgi network fusion, may not be related to contact sites [33].

In some cases, Vps13 may be present at three-way junctions between organelles, making it difficult to ascertain which membranes are being tethered based on standard fluorescent microscopy. For example, Lam6, a sterol transport protein that is anchored to the ER by an *N*-terminal transmembrane domain, has been localized at ER-mitochondria contacts, vCLAMPs and NVJs [88]. In such cases, the apparent localization to vCLAMPs could result from a 3-way junction that links ER-mitochondria and mitochondria-vacuole membranes. It is also possible that, due to its multiple membrane targeting domains, Vps13 makes contact with all three organelles at these junctions. Thus, much work is needed to determine which pairs of membranes are tethered by Vps13 at contact sites, and to identify the full set of contact sites where Vps13 functions.

## 5. Drivers of Lipid Transport

How are lipids trafficked through the Vps13 lipid transport groove? Atg2 can transport lipids bidirectionally between liposomes in vitro [42]. Whether bidirectional lipid transport occurs in vivo, and whether this is true for Vps13, has yet to be determined. For unidirectional transport, such as in membrane expansion, a lipid source, lipid sink and energy source are required. Possible energy sources for lipid transport are ATP hydrolysis-driven lipid pumps, lipid gradients formed by production or consumption of lipids in the acceptor membrane or a counter-current formed by bi-directional lipid transport of two different lipid species [2].

### 5.1. The Role of Lipid Biosynthesis

A lipid source is required for the large lipid flux needed for membrane expansion events at autophagosome or prospore membranes. Localized phospholipid synthesis at sites of membrane expansion could provide such a source. For example, the localization of lipases and phospholipid synthases at sites near Atg2 allow newly synthesized lipids to be rapidly channeled into Atg2 during autophagy [76,77,78,80]. VPS13 proteins may similarly rely on localized phospholipid synthesis at membranes to drive rapid lipid transport (Figure 3). One such membrane may be the ER, which is the major site of synthesis of bulk glycerophospholipids and cholesterol [48]. Although VPS13 proteins are not known to scaffold enzymes at MCSs, this has been reported for other MCS proteins such as Mdm1 which recruits the long chain fatty acyl-CoA synthetase Faa1 to ER-vacuole-LD three-way contact sites [71]. This may represent a widespread mechanism to facilitate lipid transport at MCSs.

### 5.2. The Role of Lipid Scramblases

Lipid acceptors on donor organelle membranes provide another mechanism for driving lipid transport. For Atg2, unidirectional lipid transport from the ER to the PAS may be driven in part by the lipid scramblase activity of its interacting partner Atg9 [89]. Atg9 is a transmembrane protein that localizes Atg2 to the PAS [47]. As a scramblase, Atg9 can translocate phospholipids between both leaflets of a membrane allowing the excess of newly transported phospholipids to move from cytosolic to luminal leaflets permitting membrane expansion [89]. Although no clear association between yeast Vps13 and a scramblase has been identified, other proteins may act as lipid acceptors. The mitochondrial adaptor Mcp1 is an integral membrane protein with two predicted heme-binding domains. Point mutations resulting in amino acid substitution within each heme-binding domain ablated the ability of overexpressed *MCP1* to rescue ERMES mutant defects without affecting Vps13 recruitment [31]. This suggests that Mcp1 has an important function in the rescue of ERMES mutants beyond simply acting as an adaptor.

Interestingly, VPS13A has recently been shown to interact with a putative scramblase, XK (Figure 3) [56]. Mouse XK homologs act as lipid scramblases to translocate phosphatidylserine (PS) across the plasma membrane leaflets [90,91]. In human cells, VPS13A and XK colocalize at ER regions that partially colocalize with mitochondria. It is interesting to speculate that VPS13A transports lipids to an acceptor membrane where they are translocated by XK to the luminal leaflet. The VPS13A-XK interaction has significant disease implications as mutations in *XK* result in McLeod syndrome, a neuroacanthocytosis syndrome with similar phenotypes to *VPS13A*-associated chorea-acanthocytosis [92]. The later age of onset for McLeod syndrome may be due to functional redundancy between XK and its 7 homologs [56]. In fact, a high-throughput proteomics study identified an interaction between VPS13A and the XK homolog, XKR2 [93] suggesting these proteins could function together.

## 6. Conclusions and Future Directions

Recent work in yeast and other model systems suggests VPS13 proteins bridge organelle membranes at MCSs to form a direct channel for lipid transport. The localization of the four human VPS13 homologs to different subsets of MCSs could explain why mutations in each are associated with distinct neurological disorders. Understanding the precise function of each homolog at specific organelles or contact sites will thus be important for understanding the cellular basis of these diverse disorders.

Additional research using the yeast model is expected to identify new interacting proteins that drive lipid transport or Vps13 membrane recruitment. Although yeast Vps13 is known to localize to a number of organelle membranes, the opposing organelles at these suspected contact sites are largely unknown. Further work is required to determine the organelle pairs tethered by Vps13 and the conditions that regulate its recruitment to each site. Characterizing the interacting partners, membrane targeting strategies and sites of function of yeast Vps13 is expected to inform future research on the human homologs and their roles in disease.

## Figures and Tables

**Figure 1 ijms-22-02905-f001:**
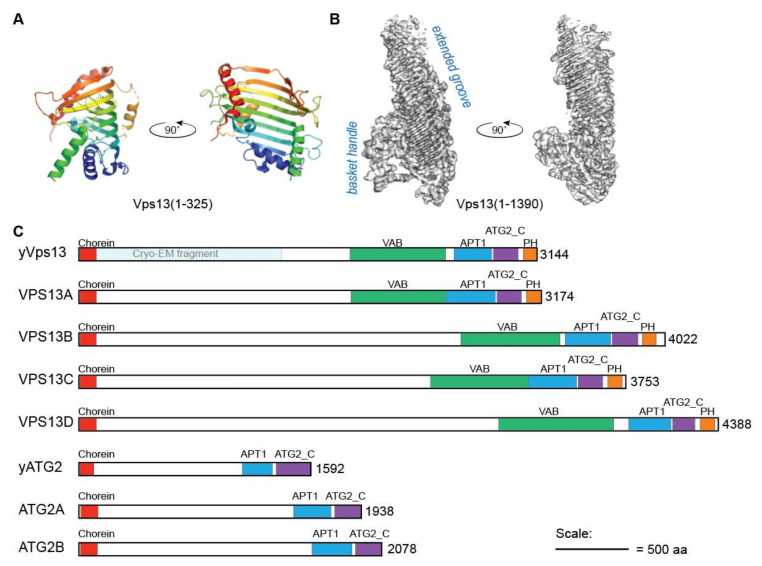
Domain architecture of VPS13 and ATG2 proteins. (**A**) Ribbon models of the crystal structure of residues 1-325 of *C. thermophilum* Vps13 lacking residues 95-132 (Protein Data Bank accession No. 6CBC) [15]. Residues are coloured blue to red from 1 to 325. (**B**) Cryo-EM structure of residues 1-1390 of *C. thermophilum* Vps13 (EMD-21113, 3.75 Å resolution) [37]. (**C**) Shared conserved domains of *S. cerevisiae* and human VPS13 and ATG2 proteins drawn approximately to scale. Domains occurring in individual proteins, such as the ubiquitin-associated (UBA) domain of VPS13D, are not shown. Domains and abbreviations are as follows: Chorein domain; VAB, Vps13 Adaptor Binding/WD40 domain; APT1 domain; ATG2_C, Autophagy-related protein 2 *C*-terminal domain; and PH, Pleckstrin homology domain. The region of yeast Vps13 corresponding to the cryo-EM structure of *C. thermophilum* Vps13 [37] is shown in pale blue.

**Figure 2 ijms-22-02905-f002:**
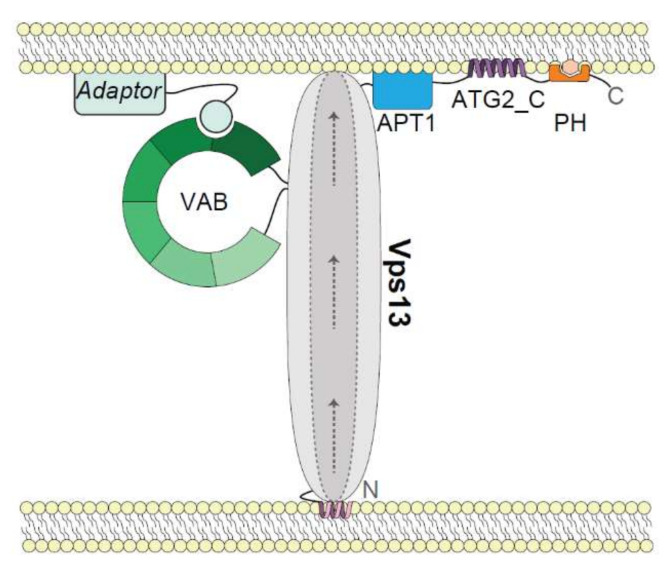
Model of Vps13 membrane targeting determinants. The Vps13 lipid transport channel bridges organelle membranes at contact sites through membrane targeting determinants on either end. An *N*-terminal helix in the chorein domain is predicted to target membranes. Within the *C*-terminus, the six-repeat Vps13 Adaptor Binding (VAB) domain binds proline-X-proline (PxP) motif-containing adaptors at a site within repeats 5–6. Additional *C-*terminal conserved APT1, Autophagy-related protein 2 *C*-terminal (ATG2_C) and Pleckstrin homology (PH) domains have putative membrane targeting roles that may function cooperatively to position the lipid transport channel. Arrows indicate the presumed direction of lipid transport along a hydrophobic groove shown by dotted lines. The *N*- and *C*-termini of Vps13 are indicated. Model not to scale.

**Figure 3 ijms-22-02905-f003:**
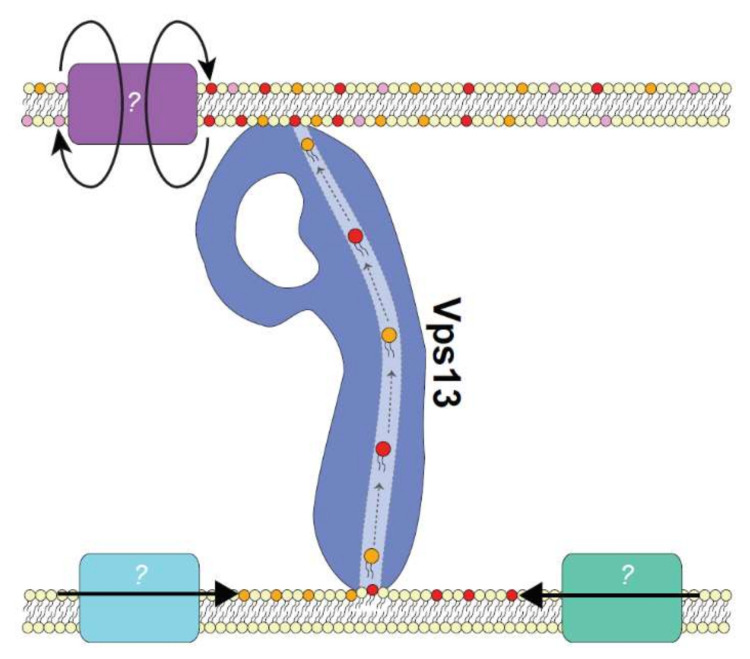
Proposed mechanism of VPS13 lipid transport at membrane contact sites. Unidirectional transport of bulk lipids by VPS13 proteins may be driven by localized lipid synthesis at donor membranes and scramblase activity on acceptor membranes. Putative lipid synthesis enzymes are shown in teal and green. These enzymes, which may or may not be coupled directly to VPS13 proteins, could provide a source of newly synthesized lipids for bulk transport. A putative scramblase at the acceptor membrane is shown in purple. Scramblases that non-specifically translocate lipids between acceptor membrane leaflets allow membrane expansion to occur. XK, a predicted phospholipid scramblase, has been recently shown to interact with VPS13A [56]. Model not to scale.

**Table 1 ijms-22-02905-t001:** Localization of human VPS13 proteins.

Homolog	Localization	Reference
VPS13A	Endoplasmic reticulum-mitochondria	[15,16,20]
	Endoplasmic reticulum-lipid droplet	[15,16]
	Mitochondria-endosome	[20]
VPS13B	Golgi apparatus	[21]
	Endosome	[17]
	Acrosome	[22]
VPS13C	Endoplasmic reticulum-endosome/lysosome	[15]
	Endoplasmic reticulum-lipid droplet	[15]
	Phagophore	[23,24,25]
VPS13D	Mitochondria	[26]
	Golgi apparatus	[26]
	Peroxisome	[26]

**Table 2 ijms-22-02905-t002:** Localization of yeast *Vps13*.

Membrane Contact Site	Reference
Nucleus-vacuole junctions (NVJs)	[18,27,28]
Mitochondria-endosome	[28]
Vacuole-mitochondria patch (vCLAMP)	[27,28]
**Single Organelle**	**Reference**
Endosome	[18,29,30]
Mitochondria	[27,28,31]
Vacuole	[18,27,28,32]
Prospore membrane	[29]
Golgi apparatus^1^	[33]
Peroxisome	[31]

^1^ Localization to the Golgi apparatus is inferred, but not directly observed.

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
