# Peer review of "The Vps13 Family of Lipid Transporters and Its Role at Membrane Contact Sites"

_ijms, 2021, doi:10.3390/ijms22062905_

Round 1
Reviewer 1 Report
In their review, "The Vps13 family of lipid transporters and its role at membrane contact sites" the authors provide an in-depth and up-to-date review of the literature with a focus on yeast Vps13. Vps13 has garnered increased interest from recent disease associations of its family members and a number of recent advances on the structure and function of Vps13 proteins and the related protein Atg2. The review is, therefore, very timely. It is also extremely well-written and impressive for its synthesis of the literature. While the manuscript is acceptable to me in its present form, there are a couple of point that would potentially increase its impact if addressed:
(1) In the introduction, as an early clinical-biological correlation it may be worth discussing that acanthocytes (spiculated RBCs), which were critical to diagnosis of neuroacanthocytosis in the pre-genetic testing era, likely form due to alterations in lipid composition of the plasma membrane.
(2) While the figures are generally of high quality, inclusion of recent structures would enhance the review and make it easier to follow the description of Vps13 structural features.
(3) Section 4 on Vps13 at contact sites would be enhanced by a figure illustrating some of the best studied examples and highlighting in particular the adaptor involved and direction of lipid transfer where known.
Author Response
Reviewer #1 Comments
In their review, "The Vps13 family of lipid transporters and its role at membrane contact sites" the authors provide an in-depth and up-to-date review of the literature with a focus on yeast Vps13. Vps13 has garnered increased interest from recent disease associations of its family members and a number of recent advances on the structure and function of Vps13 proteins and the related protein Atg2. The review is, therefore, very timely. It is also extremely well-written and impressive for its synthesis of the literature. While the manuscript is acceptable to me in its present form, there are a couple of point that would potentially increase its impact if addressed:
Response to Reviewer #1
(1) In the introduction, as an early clinical-biological correlation it may be worth discussing that acanthocytes (spiculated RBCs), which were critical to diagnosis of neuroacanthocytosis in the pre-genetic testing era, likely form due to alterations in lipid composition of the plasma membrane.
Thank you for the great suggestion. We agree that this early evidence of lipid transport defects in VPS13A mutants is both interesting and relevant, though we feel it would be more appropriate to discuss this later in the review, where we talk about erythrocytes from ChAc patients. We’ve now added this to Section 4.4.
(2) While the figures are generally of high quality, inclusion of recent structures would enhance the review and make it easier to follow the description of Vps13 structural features.
We agree that the inclusion of recent structures demonstrating the hydrophobic lipid transport groove of Vps13 would benefit readers, and have now included these structures. Note, we were not able to obtain the copyright to reuse images from the original publications within the short resubmission period and have instead used structures from public repositories.
(3) Section 4 on Vps13 at contact sites would be enhanced by a figure illustrating some of the best studied examples and highlighting in particular the adaptor involved and direction of lipid transfer where known.
The contact sites, localization determinants and direction of lipid transport at sites of Vps13 function are poorly understood. In section 4, the nucleus-vacuole junction is the only known contact site where opposing membranes are clearly defined; for all other putative contact sites occupied by Vps13 the identity of opposing organelle membranes is not well established. The direction of lipid transport has also not been determined. Although most cartoons (drawn by us and by others) show that lipids are transported away from the N-terminal chorein domain, this is purely speculative, and based largely on models of Atg2 function. Due to these challenges, we are not comfortable speculating on these points in a figure.

Reviewer 2 Report
This review on Vps13 and its role and interactions at membrane contact sites is well organized and easy to read. It mainly refers to his studies in yeast but there are references to human vsp13 and its mutations. The bibliographical references are exhaustively reported and updated.
Author Response
Reviewer #2 Comments:
This review on Vps13 and its role and interactions at membrane contact sites is well organized and easy to read. It mainly refers to his studies in yeast but there are references to human vsp13 and its mutations. The bibliographical references are exhaustively reported and updated.
Response:
We thank the reviewer for their positive comments.
Reviewer 3 Report
The article is a comprehensive review of the Vps13 family of proteins and their role in lipid transfer at the membrane contact site. Overall, the manuscript is clearly written, well described is likely to be of interest to a diverse readership, including those interested in lipid and membrane research. Although, I do not have any strong criticism for the manuscript, but have few suggestions as to the content, incorporation of which will further improve the impact of the review on the readers.
- Abbreviations should be expanded at their first appearance to help the readers.
- In section 2 and part of figure 1, I will suggest including the published structure as part of figure 1 and discuss in more detail from reference (15,25) in section 2.
- The article lacks a model figure showing the mechanism of transport based on current literature which needs to be included.
Apart from these authors need to be congratulated for their commendable effort in putting together this review article.
Author Response
Reviewer #3 Comments
The article is a comprehensive review of the Vps13 family of proteins and their role in lipid transfer at the membrane contact site. Overall, the manuscript is clearly written, well described is likely to be of interest to a diverse readership, including those interested in lipid and membrane research. Although, I do not have any strong criticism for the manuscript, but have few suggestions as to the content, incorporation of which will further improve the impact of the review on the readers.
Response to Reviewer #3:
(1) Abbreviations should be expanded at their first appearance to help the readers.
We have now expanded all abbreviations on first mention.
(2) In section 2 and part of figure 1, I will suggest including the published structure as part of figure 1 and discuss in more detail from reference (15,25) in section 2.
We have now included these structures, as requested.
(3) The article lacks a model figure showing the mechanism of transport based on current literature which needs to be included.
We have added a Figure 3 with a model demonstrating how localized lipid synthesis at donor organelle membranes and scramblase activity at acceptor membranes could drive bulk lipid transport by VPS13 at membrane contact sites.
Apart from these authors need to be congratulated for their commendable effort in putting together this review article.